# Phenolic Compounds and Oxidative Enzymes Involved in Female Fertility in Banana Plants of the Cavendish Subgroup

**DOI:** 10.3390/plants10122790

**Published:** 2021-12-16

**Authors:** Manassés dos Santos Silva, Naiala da Hora Góes, Janay Almeida dos Santos-Serejo, Claudia Fortes Ferreira, Edson Perito Amorim

**Affiliations:** 1Posgraduate Program in Biotechnology, State University of Feira de Santana, Feira de Santana 44036-900, Bahia, Brazil; manasses.tec@hotmail.com; 2Department of Agricultural Sciences, Federal University of Recôncavo da Bahia, Cruz das Almas 44380-000, Bahia, Brazil; goesdahora@gmail.com; 3Embrapa Cassava and Fruit, Cruz das Almas 44380-000, Bahia, Brazil; janay.serejo@embrapa.br (J.A.d.S.-S.); claudia.ferreira@embrapa.br (C.F.F.)

**Keywords:** floral, sterility, genetic improvement

## Abstract

The present study investigated phenolic compounds and enzymes involved in female fertility in banana plants of the Cavendish subgroup. The wild diploid Calcutta 4 and commercial cultivar Grand Naine (Cavendish subgroup) were used. The following five stages of floral development were proposed: S1 (partial vertical emission), S2 (total vertical emission), S3 (total horizontal emission), S4 (pre-anthesis), and S5 (anthesis). Following collection, pistillate (female) flowers were freeze-dried for the subsequent removal of nectaries and the analysis of phenolic compounds (PCs), antioxidant activity (DPPH and ABTS), enzymatic activity [peroxidase (POD) and polyphenol oxidase (PPO)], and total proteins (TPs). The highest values were recorded at the S3 stage, with the values decreasing as the stages progressed (until S5). At the S3 stage, the following results were obtained for Calcutta 4 and Grand Naine, respectively: PCs (32.4 and 36.1 mg GAE·g^−1^); DPPH (735.2 and 454.4 µM TE·g^−1^); ABTS (647.8 and 555.5 µM TE·g^−1^); POD (0.8 and 0.7 µmol·min^−1^·g^−1^); PPO (3.7 and 2.7 µmol·min^−1^·g^−1^); and TP (3.2 and 2.4 µmol·min^−1^·g^−1^). These results indicate that PCs and enzymes regulate female fertility, suggesting that crossbreeding should be performed from the S3 stage in cultivars of the Cavendish subgroup to achieve fruits with seeds.

## 1. Introduction

Banana farming is an agricultural activity with great economic and social relevance worldwide. Bananas are cultivated in tropical and subtropical regions, typically by small farmers. In 2019, the global annual production of bananas reached approximately 116.8 million tons in an area under the cultivation of 5.1 million hectares, rendering it the second most produced fruit in the world. In the same year, Brazil produced 6.8 million tons of bananas, becoming the fourth major producer following India, China, and Indonesia [1].

Similar to other fruit crops, banana crops are subject to production limitations due to abiotic factors, particularly water deficit and temperature extremes [2], as well as biotic factors, including various diseases, such as yellow Sigatoka (*Mycosphaerella musicola* Leach), black Sigatoka (*Pseudocercospora fijiensis*), *Fusarium* wilt (*Fusarium oxysporum* f. sp. cubense), the nematode *Radopholus similis*, and the insect *Cosmopolites sordidus* [3,4,5,6,7,8].

*Fusarium* wilt is considered a pandemic disease and among its races, the tropical race 4 stands out. This race affects the commercial plantations of cultivars of the Cavendish subgroup, which are the basis of fruit export for many countries and represent 40% of the global production [9,10]. According to Scheerer et al. [11], this race may affect 17% of the area under banana cultivation in the world in the next 20 years, leading to an estimated loss of 36 million tons, which is equivalent to USD 10 billion.

Some approaches to mitigate the adverse effects of tropical race 4 in banana plantations have been discussed, including the development of resistant cultivars through hybridization, somaclonal variation, transgenesis, cisgenesis, and gene editing [8,10,12,13,14].

Banana hybridization has limitations, particularly in cultivars of the Cavendish subgroup, including the difficulty of obtaining seeds in crossbreeding, due to the reduced fertility of this subgroup. From other triploid subgroups of banana, including ‘Prata/AAB’ and ‘Silk/AAB’, seeds can be obtained through the hybridization between commercial cultivars and wild diploids, generating progenies suitable for the selection of hybrids with agronomic and sensory characteristics in line with the market demands. From these two subgroups, several cultivars have been developed, recommended, and used by farmers [4,15,16,17,18].

Despite the limitations associated with the improvement of the Cavendish subgroup, the *Fundación Hondureña de Investigación Agrícola* (FHIA) succeeded in obtaining seeds through the crossbreeding of Cavendish cultivars, which gave rise to tetraploid mother plants, followed by subsequent crossing with improved diploids, generating triploid hybrids with commercial potential [19]. This indicates the possibility of developing cultivars of this subgroup through hybridization, despite the observed low fertility.

The low fertility in the Cavendish subgroup has been extensively studied, and the presence of a region of oxidation or necrosis has been confirmed in the septal nectary of pistillate (female) flowers, which may be associated with low fertility. Chemical compounds present in this region, which have not yet been qualified or quantified, likely inhibit pollen tube development, consequently preventing fertilization and seed formation in fruits [20,21].

According to Veitch [22], the oxidation may be formed through the accumulation of phenolic compounds (PCs) or the action of oxidative enzymes, such as peroxidase (POD) and polyphenol oxidase (PPO); these enzymes catalyze oxidation reactions based on the available PC content, preventing the oxidation or necrosis of plant tissues. Depending on the species, PCs may produce stimulatory or inhibitory effects, including the inhibition of pollen tube development [23].

To this end, we hypothesized that PCs and oxidative enzymes (e.g., POD and PPO) are involved in the oxidation or necrosis of tissues of the septal nectary of pistillate flowers in cultivars of the Cavendish subgroup, resulting in the inhibition of pollen tube development and, consequently, fertilization. To test this hypothesis, we qualified and quantified the PCs and enzymes in the septal nectaries of pistillate flowers of Grand Naine. For comparison, we used the wild diploid Calcutta 4, a universal model of genotype with high fertility.

## 2. Results and Discussion

### 2.1. Total Phenolic Compounds Content

The total phenolic compounds (PC) content varied across the different stages of floral development, ranging from 23.0 to 32.4 mg GAE·g^−1^ in Calcutta 4 and from 19.9 to 36.1 mg GAE·g^−1^ in Grand Naine (Figure 1). In both genotypes, the highest PC content was recorded at the S2 and S3 stage for Calcutta 4 (32.2 and 32,4 mg GAE·g^−1^, respectively) and S3 stage for Grand Naine (36.1 mg GAE·g^−1^).

In both genotypes, the total PC content followed a decreasing trend until the last stage S5 was reached. This finding can be justified, as at the stage, the inflorescence is in a pendulous position to the pseudostem, with open flowers that are ready to be pollinated. The total PC content of Grand Naine was higher than that of Calcutta 4.

Rocha [24] evaluated phenolics present in the distal portion of the ovary (septal nectary) of pistillate flowers of improved diploids and commercial triploids at anthesis and observed the highest total PC content in Prata-Anã (AAB, 30.8 mg GAE·g^−1^) and Grand Naine (37.5 mg GAE·g^−1^). Among the improved diploids evaluated, 089087-01 [(Malaccensis × Sinwobogi) × (Calcutta 4 × Heva)] (1.4 mg GAE·g^−1^) and TH 0301 (Terrinha-AAB × Calcutta 4) (3.5 mg GAE·g^−1^) showed a lower total PC content.

We observed differences in total PC content between diploids and triploids, indicating that these compounds play key roles in reducing fertility in bananas, such as in Grand Naine. Such sterility limits the transfer of agronomic characteristics, including disease resistance, from diploids to commercial cultivars [25].

PCs constitute a diverse group of secondary metabolites that are abundantly synthesized in the plant kingdom. They have been linked to diverse biological activities, such as antimicrobial, antioxidant, and anti-inflammatory reactions; pollinator attraction; and fruit dispersal. However, the accumulation of these compounds in the cell may be toxic, since PCs promote the production of free radicals, consequently resulting in greater oxidation and necrosis of plant tissues [22]. Based on our results, this reaction in the septal nectaries likely leads to the inhibition of pollen tube development in Grand Naine, but not in Calcutta 4, which is highly fertile, as evidenced by the higher fertilization rate of ovules.

According to Soares et al. [20], Grand Naine presents signs of oxidation or necrosis in the region of the septal nectaries of its pistillate flowers at anthesis or following pollination. Such oxidation may act as a limiting factor for fertility through the formation of a permeability barrier to pollen tube development, due possibly to the deposition of exudates derived from the female gametophyte. Consequently, this inhibition affects the development of seedless fruits, ultimately leading to sterility in some cultivars. Likewise, Fahn and Benouaiche [26] observed the disintegration of the epithelial cells and low production of nectar even before anthesis, which corresponds to the S4 stage in the present study, in the septal nectaries of pistillate flowers of cultivars in the Cavendish subgroup.

The exudates of floral nectaries are organic liquids secreted from the cell walls and membranes and comprise PCs, proteins, amino acids, carbohydrates, and terpenes [11,27]. PCs present in the exudates may also produce stimulatory or inhibitory effects, depending on the species [23].

In bananas, pollen tube behavior can be affected by the exudates present in the ovaries, resulting in a chemotropic response of an attractive or a repellent action [15]. Therefore, the PCs present in the exudates of the septal nectaries of pistillate flowers likely inhibit the growth of pollen tubes in cultivars of the Cavendish subgroup, resulting in the development of seedless fruits.

### 2.2. Antioxidant Activity

The antioxidant activity, as assessed using the DPPH radical scavenging assay, varied across the different stages of floral development of the banana genotypes, ranging from 320.3 to 735.2 µM TE·g^−1^ in Calcutta 4 and from 145.2 to 445.4 µM TE·g^−1^ in Grand Naine (Figure 2A). The highest activity was recorded at the S3 stage in both genotypes (Calcutta 4: 735.2 µM TE·g^−1^ and Grand Naine: 454.4 µM TE·g^−1^).

Based on these results, we propose a possible relationship between antioxidant activity and female fertility of the genotypes analyzed, since Calcutta 4 showed higher antioxidant activity than Grand Naine, indicating conditions conducive to increased female fertility.

Rocha [24] analyzed antioxidant activity based on DPPH scavenging in the distal portion of the ovary (septal nectary) of pistillate flowers of two improved diploids with the AA genome and the cultivars Prata-Anã (AAB) and Grand Naine (AAA) at anthesis, corresponding to the S5 stage in the present study. The author observed the highest antioxidant activity in Prata-Anã and Grand Naine (426.7 and 418.4 µM TE·g^−1^, respectively). In the improved diploids 089087-01 [(Malaccensis × Sinwobogi) × (Calcutta 4 × Heva)] and TH 0301 (Terrinha-AAB × Calcutta 4), the antioxidant activity reached 217.1 and 516.6 µM TE·g^−1^, respectively. Overall, the antioxidant activity varied between the cultivars and improved diploids.

In the present study, at the S5 stage, the oxidant activity of the DPPH radical was 278.9 µM TE·g^−1^ in Grand Naine and 413.18 µM TE·g^−1^ in Calcutta 4 (Figure 2A). These results corroborate our hypothesis regarding the association between female fertility and antioxidant activity in the septal nectaries, as evidenced by the higher antioxidant activity in Calcutta 4, which is highly fertile and can produce fruits with seeds.

We also used the ABTS (Figure 2B) radical scavenging assay and observed the highest antioxidant activity at the S3 stage of floral development. The ABTS scavenging activity varied across the different floral developmental stages, ranging from 388.0 to 647.8 µM TE·g^−1^ in Calcutta 4 and from 357.1 to 555.5 µM TE·g^−1^ in Grand Naine. Overall, higher free radical scavenging activity was observed at the S3 stage in both genotypes (Calcutta 4: 647.8 µM TE·g^−1^ and Grand Naine: 555.5 µM TE·g^−1^).

These results provide additional support to our hypothesis regarding the positive association between female fertility and antioxidant activity present in septal nectaries, as also confirmed in the analysis of antioxidant activity using the DPPH scavenging assay (Figure 2A).

PCs are present in plants as secondary metabolites and have been the focus of research in the field of phytochemistry, due to the diversity of their structure and synthesis [28]; PCs exercise biological effects that are capable of preventing or delaying free radical-mediated oxidation and ensure that the products formed after the reaction are stable [29].

The antioxidant activity can be determined by a combination of several antioxidant compounds with different mechanisms of action, as well as the interactions among them and the medium. Therefore, it is important to combine more than one method to determine the efficiency of the antioxidant system. Among the colorimetric methods, free radical scavenging assays using DPPH and ABTS stand out [30,31], and both of these were used in the present study.

The antioxidant activity based on DPPH and ABTS scavenging was strongly correlated with the total PC content at the different developmental stages of septal nectaries of pistillate flowers of Calcutta 4 and Grand Naine. Calcutta 4 showed higher antioxidant activity due to the higher total PC content than Grand Naine. The higher antioxidant activity can be attributed to the presence of substances that suppress oxidative degradation by reducing the rate of oxidation through one or more mechanisms, such as free radical inhibition [31,32].

Furthermore, the antioxidant potential depends on chemical structure, and may produce an individual effect or react synergistically with other compounds, contributing to the enhancement of several biological functions [33], for example, affecting the passage of pollen tube through the region of the nectary, inhibiting female fertility in banana cultivars of the Cavendish subgroup.

### 2.3. Enzyme Activity

The specific enzymatic activity of POD varied among the different floral development stages of the septal nectaries analyzed, ranging from 0.5 to 0.8 µmol·min^−1^·g^−1^ of protein in Calcutta 4 and from 0.3 to 0.7 µmol·min^−1^·g^−1^ in Grand Naine (Figure 3A). In both stages, higher POD activity was recorded at the S4 stage for Calcutta 4 (0.9 µmol·min^−1^·g^−1^) and S3 and S4 stage for Grand Naine (0.7 µmol·min^−1^·g^−1^). This stage corresponds to the moment when the inflorescence is completely emitted from the banana pseudostem and is in a horizontal position to the ground.

In Grand Naine, POD activity followed a decreasing trend until reaching the last stage (S5). Conversely, in Calcutta 4, POD activity followed an increasing trend until reaching the last stage (Figure 3A). Therefore, the lower POD activity in the Grand Naine cultivar, which possibly leads to the accumulation of free radicals that lead, consequently, to necrosis in the septal nectary.

The increased POD activity in Calcutta 4 diploid may be related to female fertility. An increase in activity was observed from the S3 stage onward, indicating favorable conditions for pollen tube growth and ovule fertilization.

Therefore, we propose that crossbreeding should be made from the S3 stage onward in triploid cultivars of the Cavendish subgroup to obtain fruits with seeds, since at this stage, there is no oxidation or necrosis, which occurs from the S5 stage onward. Moreover, we hypothesize that in pistillate flowers, higher antioxidant activity promotes pollen tube orientation to fertilize the ovules at the S3 stage.

According to Shepherd [34], floral developmental stage may affect fertility, since flowers that were pollinated the day before or the day after anthesis produced fewer seeds. In addition to the floral developmental stage, environmental factors, such as humidity and temperature, affect seed production [15,35,36].

POD is ubiquitous in the plant kingdom. It is partly present in the soluble form in the cytoplasm, and partly in the insoluble form on the cell wall and membrane as well as in the mitochondria [22]. POD activity is measured to study biotic and abiotic stresses, organogenesis, developmental stages, early recognition of morphogenesis during cell differentiation, plant growth, and stigma receptivity [37,38,39,40]. Due to its wide applicability in plants, POD may serve as an indicator of female fertility in bananas; as such, the higher the POD activity, the greater the female fertility, as observed in Calcutta 4 in the present study.

In addition to POD, PPO stands out as an important enzyme involved in the oxidation of phenols in plastids and their phenolic substrates in the vacuole of cells [22]. Together, POD and PPO control oxidative reactions occurring in several physiological processes, and the investigation of their enzyme activity may contribute to understanding the occurrence of oxidative necrosis in plant tissues.

In the present study, PPO activity varied across the different stages of floral development of septal nectaries, ranging from 0.7 to 3.7 µmol·min^−1^·g^−1^ in Calcutta 4 and from 1.5 to 2.7 µmol·min^−1^·g^−1^ in Grand Naine (Figure 3B). In both genotypes, higher PPO activity was recorded at the S3 stage (Calcutta 4: 3.7 µmol·min^−1^·g^−1^ and Grand Naine: 2.7 µmol·min^−1^·g^−1^). This stage corresponds to the moment when the inflorescence is completely emitted from the banana pseudostem and is positioned horizontal to the ground.

In both genotypes, PPO activity followed a decreasing trend until reaching the last stage (S5) (Figure 3B), a behavior that differed from that observed for POD activity (Figure 3A). Moreover, we observed higher PPO activity in Calcutta 4, which may be related to its higher fertility than that of the Grand Naine cultivar. This result corroborates our hypothesis that increased PPO activity is an indicator of fertile banana genotypes.

Furthermore, the protein assay was used to determine the TP content at the different developmental stages of the septal nectaries of pistillate flowers in Calcutta 4 and Grand Naine. We selected this method because it is faster, more sensitive, and subject to fewer interferences, qualities that are essential for biochemical studies aimed at the quantification of enzymes associated with their respective activities.

In both genotypes, the TP content increased according to the progression of the developmental stages, following a pattern similar to POD and PPO (Figure 4).

In the present study, the TP content through POD activity varied among the different floral developmental stages, ranging from 1.10 to 3.2 µmol·min^−1^·g^−1^ in Calcutta 4 and from 1.2 to 2.4 µmol·min^−1^·g^−1^ in Grand Naine (Figure 4A). Specifically, the TP content was higher at the S4 stage in Calcutta 4 (3.2 µmol·min^−1^·g^−1^) and at the S3 stage in Grand Naine (2.4 µmol·min^−1^·g^−1^).

Similarly, the TP content through PPO varied among the different floral developmental stages, ranging from 8.6 to 33.3 µmol·min^−1^·g^−1^ in Calcutta 4 and from 7.5 to 13.6 µmol·min^−1^·g^−1^ in Grand Naine (Figure 4B). Specifically, the TP content was higher at the S3 stage both genotypes (Calcutta 4: 33.3 µmol·min^−1^·g^−1^ and Grand Naine: 13.6 µmol·min^−1^·g^−1^). The PPO activity was higher in Calcutta 4, which shows higher fertility than Grand Naine.

Among the different developmental stages of septal nectaries, the highest TP content, through both POD and PPO activity, was recorded at the S3 stage. The PC content was also the highest at the same stage, allowing for stronger enzyme–substrate interactions and, consequently, greater enzyme activity [41].

The initial stages (S1 and S2) showed the lowest values of PC content and antioxidant activity; the activity of oxidative enzymes may be suppressed due to the availability of less substrate for interaction. Similarly, at the S4 and S5 stages, when the flowers were at pre-anthesis and anthesis, respectively, enzymatic activity was low.

PCs are the direct substrates for oxidative enzymes, acting as the modulators of the expression of enzymes, including POD and PPO [42]. These enzymes are naturally present in plant cells, and their activity is related to the PC content and oxidative potential [40]. Therefore, the investigation of the activity of these enzymes can contribute to the understanding of the occurrence of oxidation in plant tissues and its association with cellular detoxification.

The present study showed that the PC content and enzyme activity in the septal nectaries of pistillate flowers of banana may serve as the indicators of female fertility, as they produced similar effects in Calcutta 4 and Grand Naine. These findings suggest the presence of certain chemical or physical barriers to female fertility in the Cavendish subgroup.

Although meiotic abnormalities (synapses, embryo sac abortion, and chromosomal translocations, among others), genetic interactions, and physiological and environmental conditions have been proposed as the causes of infertility in bananas, there is little information on the pre-zygotic barriers that lead to the decrease in yield or even the lack of seeds [15,20,43].

Most conventional breeding programs are based on improved diploids used as the parents in crossbreeding with other diploids or some triploids that retain some female fertility.

## 3. Materials and Methods

### 3.1. Plant Material

The following two genotypes were used in the present study: the Calcutta 4 diploid, known to be fertile and produce large quantities of seeds when pollinated, and the Grand Naine cultivar of the Cavendish subgroup, which, according to the literature, is highly sterile and rarely produces seeds when used as the female parent in crossbreeding [15,18,19,20].

The genotypes were planted in May 2017 at the Embrapa Cassava and fruits (12°39′13″ S and 39°07′21″ W) in Cruz das Almas, Bahia, Brazil [44]. We used 45 plants each of Calcutta 4 and Grand Naine, both maintained in the field and managed according to the recommendations for cultivation [45].

### 3.2. Floral Development Stages

Table 1 shows the five stages of floral development proposed for the genotypes used in the present study. These stages are adapted based on the classification proposed by Fortescue and Turner [46] according to the inflorescence position. For chemical analyses, the inflorescences were collected based on these stages.

### 3.3. Sample Preparation

Pistillate (female) flowers were collected from the banana bunches of different genotypes at the five floral developmental stages described above, and stored in an ultra-freezer at −80 °C until all inflorescence stages were collected. Subsequently, nectaries were lyophilized for 72 h in a lyophilizer (Liobras-liotop L101, São Carlos, Brazil) to obtain the samples and avoid oxidation following the removal of the septal nectaries of pistillate flowers due to the action of oxidative enzymes (POD and PPO) upon contact with oxygen. Total extractable polyphenols, antioxidant activity, and enzymatic activity were analyzed with the same set of flowers from the first bunch of each plant. All analyses were performed from the collection of three nectaries per stage, with three replicates for each genotype.

### 3.4. Analysis of Total Phenolic Compounds

#### 3.4.1. Extract Preparation

Following lyophilization, the septal nectaries were removed from the flowers for further maceration in a mortar and pestle in the presence of liquid nitrogen. Samples (0.2 g) were extracted according to the method described by Larrauri et al. [47], with some modifications. Briefly, 15 mL of 50% methanol solution (methanol:water, 1:1 *v*/*v*) was homogenized in Falcon tubes, followed by immersion in an ultrasonic bath at 40 kHz for 20 min at room temperature (27 °C). The mixture was centrifuged at 11,000 rpm at 4 °C for 15 min, and the supernatant was collected. This process was repeated using 15 mL of 70% acetone solution (acetone:water, 7:3 *v*/*v*). The samples were homogenized in fresh tubes and immersed once again in an ultrasonic bath at 40 kHz for 20 min at room temperature. The mixture was centrifuged again at 11,000 rpm for 15 min, and then the collected supernatant was transferred to a 50 mL amber volumetric flask, raised with distilled water, and homogenized. Finally, the extracts were distributed in three amber flasks for freezing.

#### 3.4.2. Determination of Total Phenolic Compounds Content

The total phenolic compounds (PC) content was determined using spectrophotometry with the Folin–Ciocalteau reagent, according to the methodology described by Larrauri et al. [47]. Gallic acid (GA) was used as the standard. Absorbance was measured using a spectrophotometer at 700 nm. The GA solution [molecular weight (MW) = 170.1 Da] was prepared by dissolving 5 mg of the reagent in distilled water in a 100 mL amber volumetric flask. A calibration curve was created based on the concentrations of 0, 200, 400, 600, 800, and 1000 µL (Figure 5A). For analyses, 1 mL of the obtained extract, 1 mL of the Folin–Ciocalteau reagent (reagent:water, 1:3 *v*/*v*), 2 mL of sodium carbonate (20%), and 2 mL of distilled water were added to test tubes, and the mixture was homogenized at 25 °C. Absorbance was measured using a spectrophotometer at 700 nm in triplicate in the dark for 30 min. The blank reading was obtained on 1 mL of distilled water under the same conditions, and the results were expressed as GAE·mg·100 g^−1^. All measurements were performed in triplicate and using a control.

### 3.5. Determination of Antioxidant Activity

To determine antioxidant activity, 2,2 diphenyl-1-picrylhydrazyl (DPPH) and 2,2′-azinobis-(3-ethylbenzothiazoline-6-sulfonic acid) (ABTS) scavenging potential was measured, as proposed by Rufino et al. [48].

#### 3.5.1. DPPH Radical Scavenging Assay

For the DPPH (MW = 394.3 Da) radical scavenging assay, 24 mg of DPPH was dissolved in methyl alcohol in a 50 mL volumetric flask, followed by homogenization and measurement using a spectrophotometer at 515 nm until an absorbance of 1.1 ± 0.02 (1.080 to 1.120) was reached. Subsequently, the solution was transferred to an amber glass flask, properly labeled, and maintained at room temperature. A standard curve was created based on the 0.005 M standard solution of 6-hydroxy-2,5,7,8-tetramethylchroman-2-carboxylic acid (Trolox; MW = 250.29 Da) at concentrations of 0, 25, 50, 100, 200, 400, and 800 µL (Figure 5B). For analysis, a 150 μL aliquot of the extract was used for each sample, and 2850 μL of the DPPH solution was added to it. Absorbance was measured using a spectrophotometer at 515 nm after 1 h from the beginning of the reaction, performed in the dark. All measurements were performed in triplicate and using controls without antioxidants. The free-radical scavenging activity is expressed as the percent reduction in DPPH radical level, that is, as a function of their percent antioxidant activity (%AA), and calculated according to the following Equation (1):%AA = (Absorbance Control − Absorbance Sample)/Absorbance Control × 100(1)

#### 3.5.2. ABTS Radical Scavenging Assay

The ABTS (MW = 548.6 Da) radical scavenging activity was determined using 5 mL of 7 mM ABTS solution (192 mg) and 140 mM (378.4 mg) of potassium persulfate (MW = 270.3 Da), which were incubated at room temperature in the dark for 16 h. Thereafter, the solution was diluted with ethanol until a solution with an absorbance of 1.1 ± 0.05 (1.050 to 1.150) at 734 nm was obtained. A standard curve was created using the synthetic antioxidant Trolox at the concentrations of 0, 25, 50, 100, 200, 300, 400, and 600 μM in ethanol (Figure 5C). For analysis, a 150 μL aliquot of the extract was used for each sample, and 3000 μL of the ABTS solution was added to it. Absorbance was measured using a spectrophotometer at 734 nm after 20 min from the beginning of the reaction, which was performed in the dark. All measurements were performed in triplicate and using controls without antioxidants. The free-radical scavenging activity is expressed as the percent reduction in ABTS radical level, that is, as a function of their percent antioxidant activity (%AA), and calculated according to the following Equation (2):%AA = (Absorbance Control − Absorbance Sample)/Absorbance Control × 100(2)

### 3.6. Determination of Enzyme Activity

#### 3.6.1. POD Activity

POD (EC1.11.1.7) activity was determined according to the spectrophotometric method described by Allain et al. [49] and modified by Lima [50]. To prepare the extract, 0.3 g of the frozen sample was mixed in 0.2 M phosphate buffer (pH 6.7). Polyvinylpyrrolidone (PVP) was added at 2 g·g^−1^ of the sample (a pinch) to inhibit PCs and oxidation. The samples were centrifuged at 4000 rpm for 10 min at 4 °C, and the supernatants were transferred to Eppendorf tubes and frozen. To determine POD activity, 0.5 mL of hydrogen peroxide (H_2_O_2_) solution, 0.5 mL of phenol solution, and 4-aminoantipyrine (AAP) were used per 1 mL of the supernatant. The tubes containing the reaction system were maintained in a water bath for 5 min, and the reaction was interrupted by adding 2 mL of absolute ethanol. Absorbance was measured at 505 nm. The specific activity of POD is expressed as µmol of H_2_O_2_ decomposed per minute per milligram of protein. All measurements were performed in triplicate and using controls.

#### 3.6.2. PPO Activity

PPO (EC1.14.18.1, monophenol mono-oxygenase and EC1.10.3.2, o-diphenol: oxidoreductases) activity was determined according to the spectrophotometric method described by Allain et al. [49] and modified by Cano et al. [51]. To prepare the extract, 0.3 g of the sample was mixed with 0.05 M phosphate buffer (pH 6.0). PVA was added at 2 g·g^−1^ of the sample (a pinch) to inhibit PCs. The samples were centrifuged at 4000 rpm for 10 min at 4 °C, and the supernatants were transferred to Eppendorf tubes and frozen. To determine PPO activity, 0.5 mL of the extract, 0.8 mL of sodium phosphate buffer solution, and 50 mL of 0.01 M catechol solution were used, and the resulting solution was incubated in a water bath for 30 min at 30 °C. To stop the reaction, 0.8 mL of 2 M perchloric acid was added. Absorbance was measured using a spectrophotometer at 395 nm. The PPO activity was defined as the variation in one absorbance unit per μmol of transformed catechol per minute per milligram. All measurements were performed in triplicate and using controls.

### 3.7. Determination of Total Proteins

Total protein (TP) content at different developmental stages of the septal nectaries of pistillate flowers of banana was determined using a microassay with the commercial kit from Bio-Rad, based on the Bradford [52] method, for 15 min in the dark. A standard curve was created with casein (CAS: 9000-71-9) at concentrations of 0, 10, 20, 30, 40, and 60 μg (Figure 5D). Absorbance was measured using a spectrophotometer at 595 nm. Protein content is directly proportional to absorbance, since the product formed by the reaction between the protein and the dye present in the Bradford protein assay reagent shows a strong signal at 595 nm. The protein content is expressed as milligram of protein per gram of sample.

### 3.8. Statistical Analysis

The experimental design was completely randomized with a 2 × 5 factorial scheme (genotypes × floral development stages). The variables evaluated were subjected to analysis using a polynomial regression model. All analyses were performed using R [53].

## 4. Conclusions

Based on our results, the following questions can be raised for further research: Could some stimulatory or inhibitory effects on pollen tubes in the region of septal nectaries in bananas be attributed to PCs and oxidative enzymes (e.g., POD and PPO)? Is higher PC content related to higher POD and PPO activity in septal nectaries, thus enabling fertilization? Do enzymes secreted by the style promote pollen tube growth before reaching the nectary to promote egg fertilization?

These questions must be addressed to fill the knowledge gap regarding the components present in septal nectaries of pistillate flowers of bananas. Moreover, elucidating the genes, proteins, and other molecules encoding these components may allow for the recognition and characterization of biochemical pathways related to female fertility in cultivars of the Cavendish subgroup. The acquired knowledge will also serve as the foundation for developing strategies to overcome the sterility crisis in bananas, contributing to the development of new cultivars.

To circumvent the current limitations, we suggest some modifications when performing crossbreeding in Grand Naine, based on the results obtained in the present study. Specifically, we recommend performing crossbreeding from the S3 stage onward, when the plant bears the inflorescence completely emitted from the pseudostem and positioned horizontally to the ground. Another potential strategy is to treat the plants with antioxidant compounds and/or hormones before pollination to protect the pollen at the time of crossbreeding. Finally, characterizing the proteomic profile of proteins secreted in the nectary, with possible links to necrosis and pollen tube inhibition, at the S4 (pre-anthesis) and S5 (post-anthesis) stages may also help elucidate the compounds responsible for the low fertility observed in cultivars of the Cavendish subgroup.

## Figures and Tables

**Figure 1 plants-10-02790-f001:**
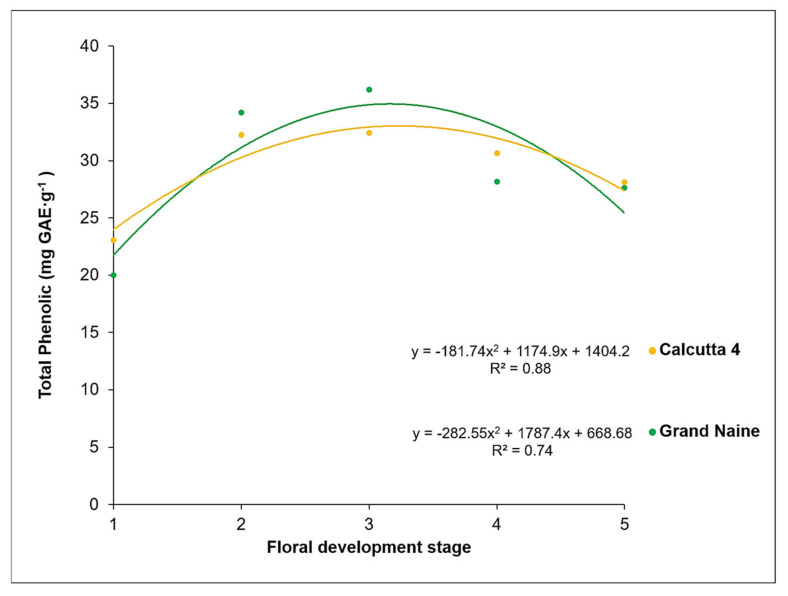
Total phenolic content at different floral developmental stages of the nectary of Calcutta 4 and Grand Naine.

**Figure 2 plants-10-02790-f002:**
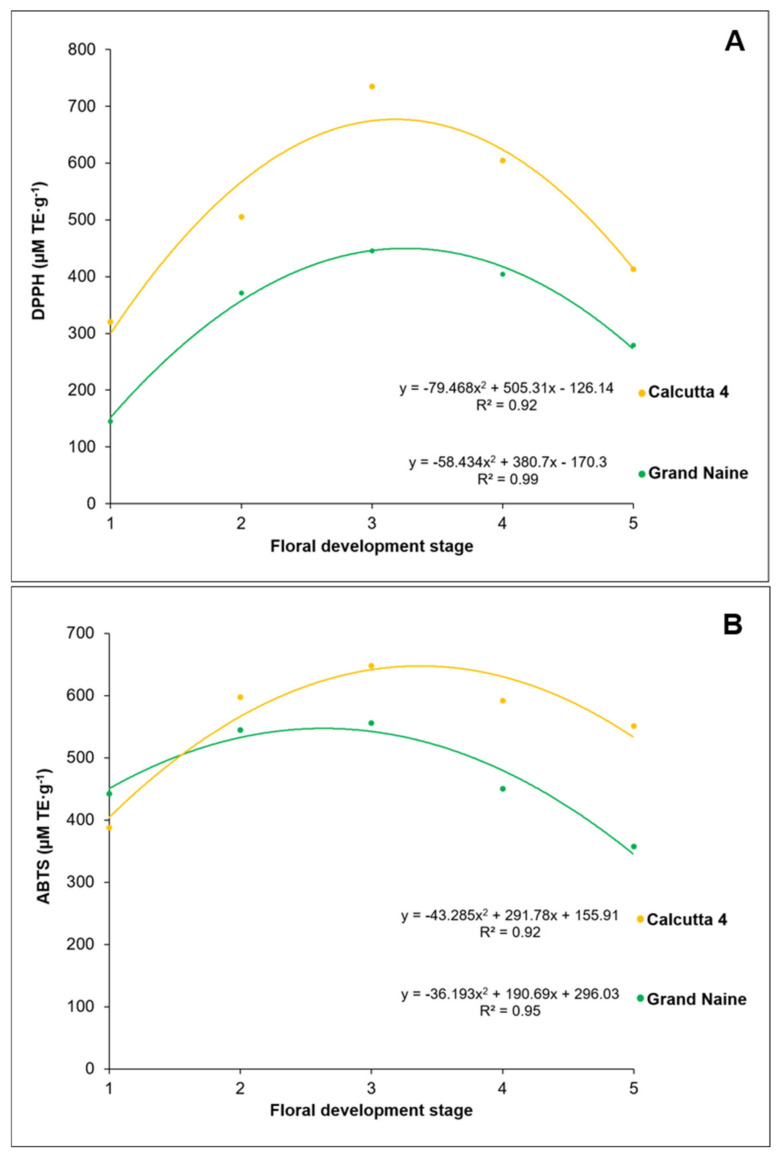
2,2-Diphenyl-1-picrylhydrazyl (DPPH) (**A**) and 2,2′-azinobis-(3-ethylbenzothiazoline-6-sulfonic acid) (ABTS) (**B**) radical scavenging activity at different stages of floral development of the nectary in Calcutta 4 and Grand Naine.

**Figure 3 plants-10-02790-f003:**
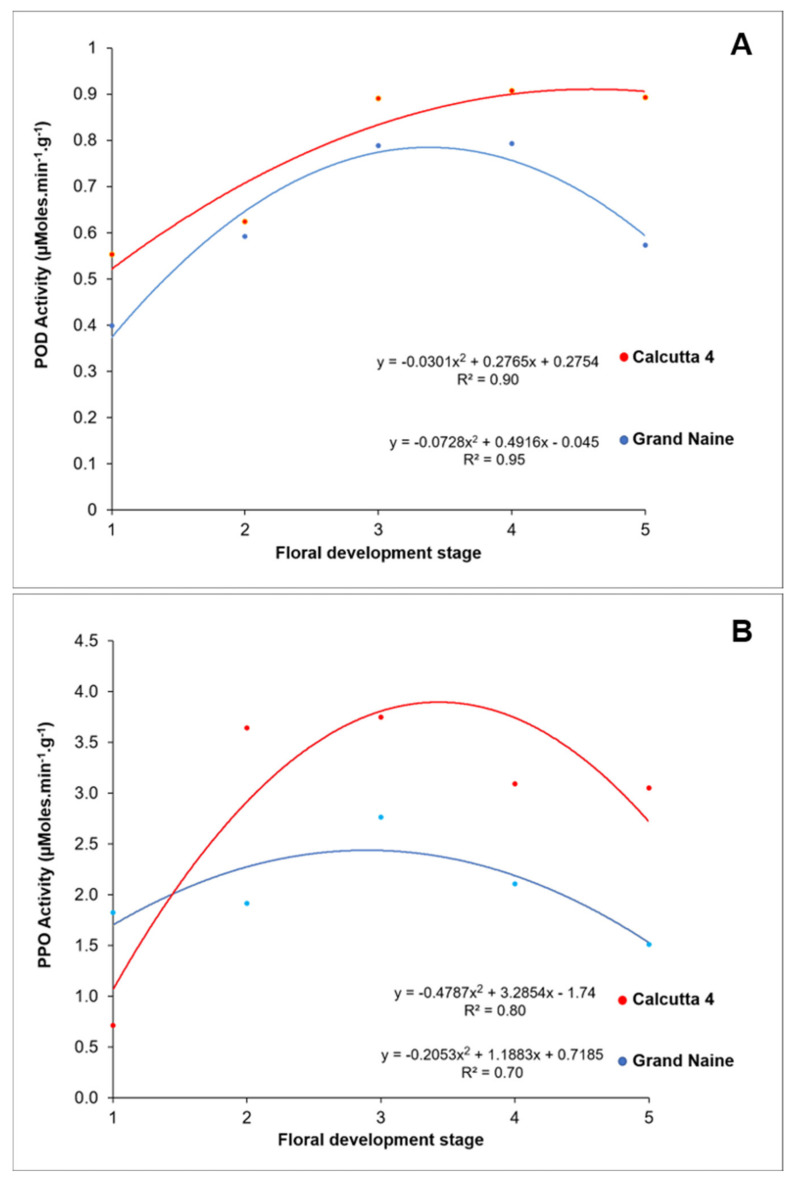
Specific activity of peroxidase (POD) (**A**) and polyphenol oxidase (PPO) (**B**) at different stages of floral development in Calcutta 4 and Grand Naine.

**Figure 4 plants-10-02790-f004:**
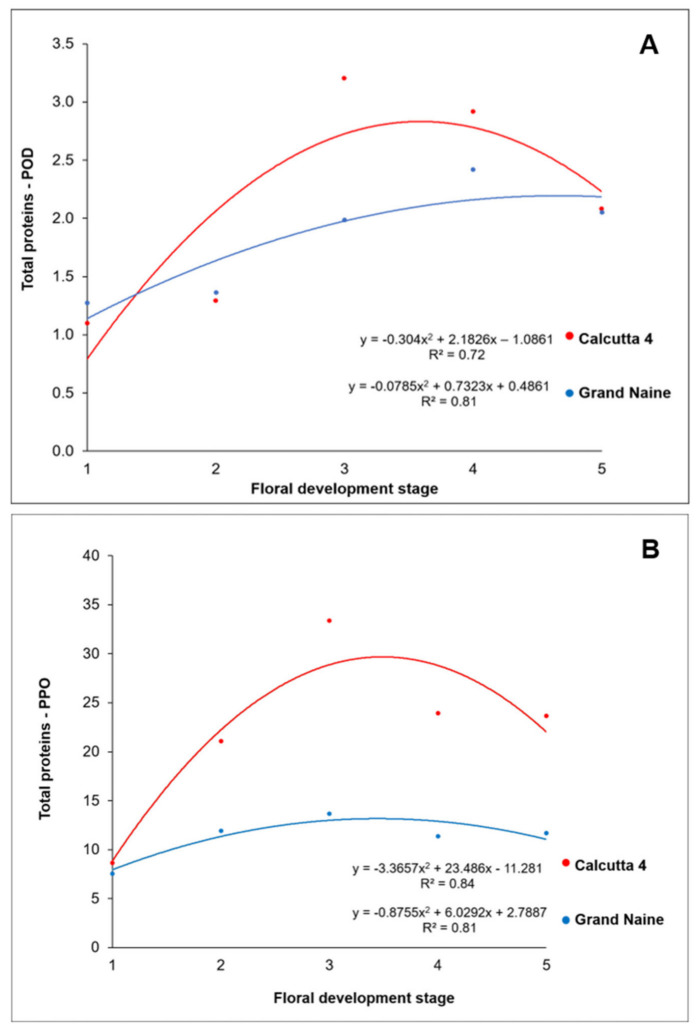
Protein content assay through the enzymatic activity of (**A**) peroxidase (POD) and (**B**) polyphenol oxidase (PPO) per gram of protein at different stages of floral development in Calcutta 4 and Grand Naine.

**Figure 5 plants-10-02790-f005:**
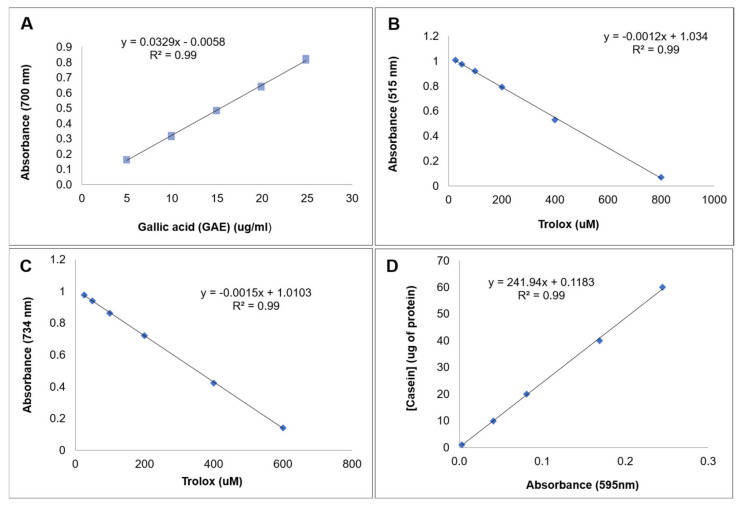
Gallic acid standard curves (GA) for the determination of total phenolic compounds (**A**); Trolox (6-hydroxy-2,5,7,8-tetramethylchromo-2-carboxylic acid) for the determination of DPPH (2.2-diphenyl-1-picryl-hydrazyl) (**B**) and ABTS (2,2′azinobis-[3-ethylbenzthiazoline-6-sulfonic acid]) (**C**); and Casein (CAS: 9000-71-9) by the method of Bradford (1976) for quantification of total proteins (**D**).

**Table 1 plants-10-02790-t001:** Proposed developmental stages of pistillate (female) flowers (A–E).

Acronym	Stage	Description	
S1	Stage 1	Partial emission of the pseudostem inflorescence—vertical position (without viewing the base of the inflorescence)	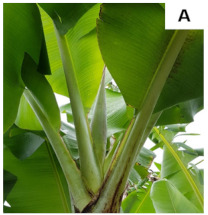
S2	Stage 2	Total emission of the pseudostem inflorescence—vertical position (base of visible inflorescence)	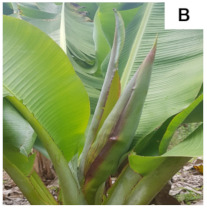
S3	Stage 3	Total inflorescence emission—horizontal position to the ground	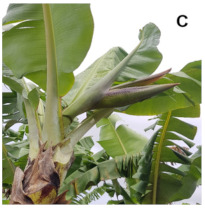
S4	Stage 4	Inflorescence in pendular position to the pseudostem with closed flowers (Pre-anthesis)	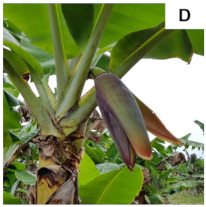
S5	Stage 5	Inflorescence in pendular position to the pseudostem with open flowers (Anthesis)	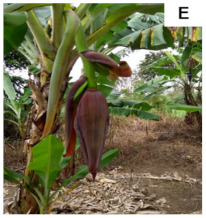

## Data Availability

All data were found in Embrapa Mandioca e Fruticultura, Brazil.

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
