# Peer review of "Phenolic Compounds and Oxidative Enzymes Involved in Female Fertility in Banana Plants of the Cavendish Subgroup"

_plants, 2021, doi:10.3390/plants10122790_

Round 1

Reviewer 1 Report

The authors re-submitted a publication on the cultivation and reproduction of bananas. I am impressed with the work the authors put into improving the article. As it stands, the results are better presented, the discussion is well conducted, the methods are described correctly. Conclusions were also formulated. The authors took into account the previous tips on how to improve the publication. Now the work is more coherent and understandable. The presentation of the results has been improved and describes the observed phenomenon well. The authors also added a Conclusions section in which they took into account the reviewer's suggestions. Currently, the publication meets the publication requirements and may be published in Plants.

Author Response

Point: The authors re-submitted a publication on the cultivation and reproduction of bananas. I am impressed with the work the authors put into improving the article. As it stands, the results are better presented, the discussion is well conducted, the methods are described correctly. Conclusions were also formulated. The authors took into account the previous tips on how to improve the publication. Now the work is more coherent and understandable. The presentation of the results has been improved and describes the observed phenomenon well. The authors also added a Conclusions section in which they took into account the reviewer's suggestions. Currently, the publication meets the publication requirements and may be published in Plants.

Response: The authors are grateful for their considerations.

Reviewer 2 Report

The author of the manuscript entitled “Phenolic compounds and oxidative enzymes involved in female 2 fertility in banana plants of the Cavendish subgroup” did good research work. There are many issues where the author needs to concentrate and revise prior to its acceptance and publication. Some of the issues are as follows:

Abstract:

Introduction:

Line 76:  Which region? Explain in the following.

“According to Veitch [22], this region of oxidation may be formed through…”

Figure 1: Remove the memo from fig.

Line 93: Fig 1 showed that PC is higher at stage 2 of the culcutta4 cultivar.

The statement is different in the text.

“In both genotypes, the highest PC content was recorded at the S3 stage”. Explain

Line 184-193:Author need to revise these sentences. Instead of keeping unnecessary sentences, discuss the role of phenolic compounds on plant fertility. Give mechanism with recent citations.

The author need to estimate the phenolic compounds composition, concentrations and their role in banana fertility. Determining TPC according to the methodology described by Larrauri, Rupérez, and Sauracalixto is not enough to draw conclusions. Use either HPLC, GC-MS or QTOF methods to estimate PC in the sample. It will be easy for you to discuss the results with these data. 

Line 203: “Such callsign is related”. “Such” stands for?

Line 211: Check the figure. The following statement is wrong.

 “higher POD activity was recorded at the S3 stage (Calcutta 4: 0.89 µmol·min-1 ·g-1 and Grand Naine: 0.78 µmol·min-1 ·g-1 ).

Line 233 to 240: Not related to your experiment. Keep suitable literature that supports your hypothesis and results.

Line 242-249: Revise it or remove it.

The author of the research paper needs to revise and rewrite the result and discussion section of the paper. Important citations, phenolic compounds data, correlations between antioxidant compounds and fertility need to be included in the text.

Author Response

The author of the manuscript entitled “Phenolic compounds and oxidative enzymes involved in female 2 fertility in banana plants of the Cavendish subgroup” did good research work. There are many issues where the author needs to concentrate and revise prior to its acceptance and publication. Some of the issues are as follows:

Abstract:

Point 1: Introduction: Line 76:  Which region? Explain in the following. “According to Veitch [22], this region of oxidation may be formed through…”

Response 1: Ajusted.

Point 2: Figure 1: Remove the memo from fig.

Response 2: Ajusted.

Point 3: Line 93: Fig 1 showed that PC is higher at stage 2 of the culcutta4 cultivar. The statement is different in the text.“In both genotypes, the highest PC content was recorded at the S3 stage”. Explain

Response 3: Ajusted. In both genotypes, the highest PC content was recorded at the S2 and S3 stage for Calcutta 4 (32.2 and 32,4 mg GAE·g-1, respectively) and S3 stage for Grand Naine (36.1 mg GAE·g-1).

Point 4: Line 184-193:Author need to revise these sentences. Instead of keeping unnecessary sentences, discuss the role of phenolic compounds on plant fertility. Give mechanism with recent citations.

Response 4: Due to the lack of literature related to the topic of this study.

Point 5: The author need to estimate the phenolic compounds composition, concentrations and their role in banana fertility. Determining TPC according to the methodology described by Larrauri, Rupérez, and Sauracalixto is not enough to draw conclusions. Use either HPLC, GC-MS or QTOF methods to estimate PC in the sample. It will be easy for you to discuss the results with these data. 

Response 5: We agree with the reviewer´s comment, however, our results should not be disregarded due to lack of the HPLC analysis. Our work aimed to present the scientific community new knowledge about banana fertility, a not very well understood, and limiting factor in bananas. It does not propose in any way to liquidate the topic, but to add information to the information that already exists and allow for the exploration for broadening new future perspectives, including HPLC analysis. We in the near future will be using HLPC in samplings of the same stages of floral development presented here in this manuscript. Furthermore, proteomic analysis of the latter samplings is undergoing as part of a thesis of a Doctoral fellowship to increase knowledge of the referred topic. Therefore, we hope the reviewer will take into consideration of the importance of the results presented in this manuscript regarding banana fertility, which certainly will contribute to the topic for bananas of the Cavendish subgroup.

Point 6: Line 203: “Such callsign is related”. “Such” stands for?

Response 6: Ajusted,

Point 7: Line 211: Check the figure. The following statement is wrong.  “higher POD activity was recorded at the S3 stage (Calcutta 4: 0.89 µmol·min-1 ·g-1 and Grand Naine: 0.78 µmol·min-1 ·g-1).

Response 7: Ajusted.

Point 8: Line 233 to 240: Not related to your experiment. Keep suitable literature that supports your hypothesis and results.

Response 8: Due to the lack of literature related to the topic of this study, we suggest strategies that encourage further studies, such as, for example, the performance of crossings from the S3 stage.

Point 9: Line 242-249: Revise it or remove it.

Response 9: Ajusted.

Point 10: The author of the research paper needs to revise and rewrite the result and discussion section of the paper. Important citations, phenolic compounds data, correlations between antioxidant compounds and fertility need to be included in the text.

Response 10: Ajusted.

Reviewer 3 Report

Review plant-1466048 M. dos Santos Silva, N. da Hora Góes, J. A. dos Santos-Serejo, C. Fortes Ferreira, E. Perito Amorim “Phenolic compounds and oxidative enzymes involved in female fertility in banana plants of the Cavendish subgroup”

The authors present interesting results on phenolic compounds and oxidative enzymes (peroxidase and polyphenol oxidase) in five stages of floral development in the banana plant of the Cavendish subgroup. The highest values were seen for the stage 3. Phenolic compounds and oxidative enzymes were shown as important factors for regulation of fertility. Thus, the authors present interesting results with impact on scientists as well as on plant breeder.

However, there are some points to be corrected to improve the quality of the manuscript:

Line 29: one of the keywords is already mentioned in the title; please look for an alternative
Lines 92-95: please give all values only with one digit after decimal point
Line 152: please change “3- ethylbenzo” into “3-ethylbenzo”
Line 378: please change “were expressed” into “are expressed”
Line 399: please change “was expressed” into “is expressed”
Line 401: please change “formula (2)” into “formula (1)”
Lines 416/434: please change twice “was expressed” into “is expressed”
Line 460: please change “was expressed” into “is expressed”

Author Response

Review plant-1466048 M. dos Santos Silva, N. da Hora Góes, J. A. dos Santos-Serejo, C. Fortes Ferreira, E. Perito Amorim “Phenolic compounds and oxidative enzymes involved in female fertility in banana plants of the Cavendish subgroup”

The authors present interesting results on phenolic compounds and oxidative enzymes (peroxidase and polyphenol oxidase) in five stages of floral development in the banana plant of the Cavendish subgroup. The highest values were seen for the stage 3. Phenolic compounds and oxidative enzymes were shown as important factors for regulation of fertility. Thus, the authors present interesting results with impact on scientists as well as on plant breeder.

However, there are some points to be corrected to improve the quality of the manuscript:

Point 1: Line 29: one of the keywords is already mentioned in the title; please look for an alternative

 Response 1: Ajusted.

Point 2: Lines 92-95: please give all values only with one digit after decimal point

 Response 2: Ajusted.

Point 3: Line 152: please change “3- ethylbenzo” into “3-ethylbenzo”

 Response 3: Ajusted.

Point 4:Line 378: please change “were expressed” into “are expressed”

 Response 4: Ajusted.

Point 5:Line 399: please change “was expressed” into “is expressed”

 Response 5: Ajusted.

Point 6:Line 401: please change “formula (2)” into “formula (1)”

 Response 6: Ajusted.

Point 7:Lines 416/434: please change twice “was expressed” into “is expressed”

 Response 7: Ajusted.

Point 8: Line 460: please change “was expressed” into “is expressed”

 Response 8: Ajusted.

Round 2

Reviewer 2 Report

The author of the manuscript responded to all the queries and revised the manuscript. The manuscript can be accepted.

This manuscript is a resubmission of an earlier submission. The following is a list of the peer review reports and author responses from that submission.

Round 1

Reviewer 1 Report

The major problem of this manuscript is simplicity of methods. Authors performed only spectrophotometric analysis without conducting HPLC analysis of phenolic compounds. Results of HPLC would be of much higher importance, in order to obtain change of phenolics profile over the stages. It is known that Folin–Ciocalteau reagent can react in different extant with different phenolic compounds, and also this reagent can reagent with proteins. So results based only on this method can be misleading.

Some technical mistakes:

Why in the abstract authors use =, write the sentences.

Radicals are written wrongly, check literature for correct way.

In figures there are no SD. Units should be written in the same manner as in text.

English have to be improved.

Reviewer 2 Report

The paper can be published in Plants after the minor revision. In my opinion it is an very interesting study. The presented manuscript gives very interesting results concerning the reproduction of bananas. All references are cited in the text. The experiment was well planned. However, paper requires some minor corrections. To improve the quality of the presentation of your work, please consider the following suggestions:

  1. Some sentences are incomprehensible to me. This is especially true for lines 176-180 and 194-196. Please recompose them.
  2. Although the authors stated in the Materials and Methods that they used statistical methods, this is completely absent from the Results and Discussion. Please take into account the statistics in the results and mark the statistically significant differences. This applies to all the results presented, especially when the results for different varieties are compared.
  3. In my opinion, the last sentences (lines 307-329) in Result and Discussion should be changed into a separate chapter - Conclusions. Please consider rewriting the text in this way and include it in the Conclusions a brief and concise explanation of the observed phenomenon, and proposals for further research.

Reviewer 3 Report

The manuscript "Phenolic compounds and oxidative enzymes involved in female fertility in banana plants of the Cavendish subgroup" is interesting and very important for plant breeding. However, there are some issues that need to be addressed prior to its publication. For instance ----

Introduction

Line 41: the following sentence is not connected and is unnecessary..

“Fusarium wilt is considered a pandemic disease due to its occurrence on all continents.”

Line 92:  Figure 5 is not in chronological order.

Line 94:  which findings you are talking about?’’’

Estimation of total phenolic contents (PC)by using Folin–Ciocalteau reagent is not sufficient to determine and estimate the PC present in the sample.

I suggest the author to do HPLC analysis for PC.

.Line 376: write the make, model, address (city and country) of the instrument     

Line 397: How much was the temperature?

Line 400: R2 stand for “straight line”? Write properly.

Line 407: The formula used in the calculation of DPPH is missing in the text.